# Hepatoprotective Effects of Sweet Cherry Extracts (cv. Saco)

**DOI:** 10.3390/foods10112623

**Published:** 2021-10-29

**Authors:** Ana C. Gonçalves, José D. Flores-Félix, Ana R. Costa, Amílcar Falcão, Gilberto Alves, Luís R. Silva

**Affiliations:** 1CICS–UBI—Health Sciences Research Centre, University of Beira Interior, 6201-506 Covilhã, Portugal; anacarolinagoncalves@sapo.pt (A.C.G.); jdflores@usal.es (J.D.F.-F.); anarfcosta1990@gmail.com (A.R.C.); gilberto@fcsaude.ubi.pt (G.A.); 2CIBIT—Coimbra Institute for Biomedical Imaging and Translational Research, University of Coimbra, 3000-548 Coimbra, Portugal; acfalcao@ff.uc.pt; 3Laboratory of Pharmacology, Faculty of Pharmacy, University of Coimbra, 3000-548 Coimbra, Portugal; 4CPIRN-UDI/IPG—Centro de Potencial e Inovação em Recursos Naturais, Unidade de Investigação para o Desenvolvimento do Interior do Instituto Politécnico da Guarda, 6300-559 Guarda, Portugal

**Keywords:** antioxidant effects, cytotoxicity, molecular docking, HepG2 cells, sweet cherry phenolics

## Abstract

Cancer is the second cause of death worldwide. Among cancers, hepatocellular carcinoma is one of the most prevalent. Evidence indicates that the daily consumption of fruits and vegetables can prevent the onset of various cancers due to the presence of bioactive compounds. Sweet cherries are known for their richness in phenolics, including anthocyanins, which are the major constituents, and presumably, the key contributors to their biological activity. Therefore, the present study aimed to evaluate the effects of three different cherry fractions on human hepatocellular carcinoma (HepG2) cells viability and effectiveness to improve the redox status of these cells under oxidative damage induced by nitric oxide radicals and hydrogen peroxide. Phenolic characterization of fractions was performed by Fourier transform infrared spectroscopy. The obtained results indicated that enriched phenolic fractions of sweet cherries (cv. Saco, can impair cell viability and suppress cells growth after 72 h of exposure, promoting necrosis at the highest tested concentrations (>50 µg/mL). Additionally, fractions also showed the capacity to protect these cells against oxidative injury by capturing radicals before they can attack cells’ membrane and by modulating reactive oxygen and nitrogen species generation, as demonstrated by bioinformatic tools.

## 1. Introduction

Cancer is considered one of the most alarming medical problems worldwide. This disease results from genetic and epigenetic alterations of oncogenes or tumour suppressor genes, leading to uncontrolled and rapid cells growth, proliferation and invasion to other organs and tissues [1]. Chemotherapy is frequently used to heal this malignancy, however this treatment presents few tumour-selectivity drug delivery and can cause several side effects, such as anaemia, emotional distress, depression, fatigue, vomits, pain, among others [2,3,4]. Hepatocellular carcinoma is one of the most lethal cancers. Indeed, in the last year, it was responsible for killing approximately 830,000 individuals [1]. It is mainly caused by hepatitis B or C virus infection, cirrhosis, non-alcoholic fatty liver disease, alcohol-induced liver disease, and exposure to aflatoxins, nitrosamines and/or other carcinogens [5]. When detected at an early-stage, patients can be subjected to curative surgical treatments, such as tumour resection and ablation. However, in the majority of the cases, its diagnosis is difficult and only occurs at an advanced stage, resulting in death [3]. In fact, this cancer type presents an overall 5-year survival rate very low (below 9%) [1,5]. Current palliative treatment medicine includes the administration of sorafenib, which is a multi-kinase inhibitor for systemic chemotherapy. Nevertheless, this one is very expensive, displays serious side effects, namely at cardiovascular level, and only increases survival rates for about 2.5 months [3,6].

Therefore, there exists an urgent demand for developing novel, more efficient, specific, and safer chemopreventive, chemotherapeutic and/or adjuvant agents to fight against cancer. Since several epidemiological and animal studies highlight that daily consumption of vegetables and fruits are robust and viable strategies useful to reduce the development and/or to act as a complementary treatment against various cancer types, being capable of increasing survival rates [7,8,9,10], it is not surprising the increasing incorporation of molecules derived from natural products in antitumor drugs [7]. 

Among these molecules, phenolic compounds, such as phenolic acids, flavan-3-ols and anthocyanins, have been largely studied due to their antioxidant, anti-inflammatory and antimutagenic effects, as well as capacities to inhibit cell development and differentiation, modulate cell metabolism, induce cell cycle arrest and apoptosis, and control the expression of genes involved on cancer cells growth, among others [3,11,12,13,14]. Thus, natural products rich in these bioactive compounds, such as sweet cherries, have been a target of extensive and deeper studies in order to discover their full biological potential and safe dosage. Sweet cherries (*Prunus avium* Linnaeus) are perishable and colourful fruits that belong to the Rosaceae family [15]. They are highly attractive and appreciated owing to their taste, aroma and colour, as well as due to their nutritional values and functional properties, once they present considerable levels of phenolics, including anthocyanins (0.61 to 108.5 mg cyanidin 3-*O*-rutinoside equivalent per 100 g of fresh weight), which are the main responsible for their red colour and considered the key contributors for their noticeable antioxidant effects and ability to mitigate many pathological conditions, including cancer [12,16,17,18,19].

In fact, the health benefits of *Prunus avium* plant parts are known since ancient times, where they were used to treat several ailments, such as ague, bellyache, diarrhea, jaundice and tonsillitis conditions and urogenital, urinary and intestinal disorders [20]. Focusing on their fruits, they are preferentially consumed fresh, but they also can be dried (with or without sugar residues) or processed into juices, jam, jellies, beverages, being, hence available all year in the market [21]. Given the crescent interest in this fruit, its cultivation is increasing worldwide. According to Faostat, in 2019, around 3,600,000 tons of sweet cherries were produced globally (which corresponds to an increase by 60% comparatively to 10 years before), being Turkey (628,000 tons), the United States of America (398,000 tons) and Iran (140,000 tons) the biggest producers. 

Considering the high liver cancer death rates and the lack of reports about the effect of sweet cherries on tumoral hepatic cell proliferation and cytotoxicity, the present study provides the first information about the pharmacological effects of sweet cherry concentrated phenolic-rich fractions (cv. Saco) on inhibiting the growth of human hepatocarcinoma (HepG2) cells. This cell line was the selected one once it presents many morphological characteristics of liver parenchymal cells and enzymes responsible for the activation of various xenobiotics, being, therefore, a widely used in vitro model for human liver cancer research.

Additionally, the cytoprotective effects and mechanism of action of the fractions against oxidative damage induced by nitric oxide (NO) and hydrogen peroxide (H_2_O_2_) on HepG2 cells proliferation, apoptosis, migration and invasion were also explored herein for the first time. Additionally, the phenolic profile was characterized using the Fourier transform infrared (FT-IR) spectroscopy.

## 2. Materials and Methods

### 2.1. Chemical Reagents

All chemicals used were of analytical grade. N-(1-Naphthyl)ethylenediamine dihydrochloride, sulfanilamide, and sodium nitroprusside dihydrate (SNP) was obtained from Alfa Aesar (Karlsruhe, Germany). The remaining were purchased from Sigma-Aldrich (St. Louis, MO, USA). Water was deionized using a Milli-Q water purification system (Millipore Ibérica, Madrid, Spain).

### 2.2. Cherry Samples

Approximately 1 kg of Saco sweet cherry from Fundão region (Portugal) was collected manually at the commercial stage in June of 2017 and provided by Cerfundão, a local company. The fruits were immediately transported to the laboratory facilities, where pits were removed and separated from the pulp. Then, cherries’ pulp was frozen with liquid nitrogen and maintained at −80 °C until lyophilization. After lyophilization, cherry fruit was powdered and divided into three aliquots, which, in turn, were used for the preparation of the extracts. Saco was the cultivar chosen since this one already proved to have notable health-promoting properties, including antioxidant and anti-cancer activities on human colon HT29 and Caco-2 cancer cells [12,19,22,23,24,25,26].

### 2.3. Preparation of Crude Cherry Extracts

One gram of dried cherry pulp powder was homogenised with 20 mL ethanol:water (70:30, *v/v*) and subjected to 2 h stirring in the dark at room temperature. Then, the extract was centrifuged at 2900× *g* for 10 min. The supernatant was separated and evaporated under reduced pressure at 37 °C. Finally, the resulting extract was dissolved in 50 mL deionised water and stored at −20 °C to avoid phenolic degradation until purification.

#### Preparation of Phenolic Concentrated Cherry Extracts

In order to obtain the phenolic-rich extracts, a solid-phase extraction (SPE) procedure was performed using a Sep-Pak C18 column (70 mL/10,000 mg) from Macherey-Nagel (Düren, Germany). The purification is presented in Figure 1. Briefly, the aqueous cherry extract was placed into the column preconditioned with 20 mL ethyl acetate, 20 mL ethanol and 20 mL 0.01 mol/L HCl. A fraction containing non-coloured phenolics was eluted with 20 mL ethyl acetate (fraction I). A second fraction with anthocyanins (fraction II) was eluted with 40 mL ethanol acidified with 0.1% HCl to prevent anthocyanins degradation. To obtain fraction III (total extract), another C18 solid-phase extraction was performed, being this one eluted with 40 mL ethanol containing 0.1% HCl. Finally, each fraction was evaporated to complete dryness. The obtained residues were resolubilized in 5 mL deionized water and lyophilized. After that, they were stored in silica protected from light until analyses. Extractions were carried out in triplicate.

### 2.4. Fourier Transform Infrared (FT-IR) Spectrometry Analysis

Functional groups of cherry fractions were investigated using a Nicolet Is10 FT-IR spectrometer (Thermo Scientific, Waltham, MA, USA) equipped with a diamond total reflectance accessory (ATR) with a zinc selenide crystal. Briefly, lyophilized and powdered samples were applied on ATR, compressed, and pressed under vacuum placed in the sample holder of the equipment to maximize the surface of contact. All fractions were analysed three times in the same conditions. The background was collected under identical conditions to the samples and then subtracted from the sample spectra. The software OMNIC version 8.3 (Thermo Nicolet Co., Waltham, MA, USA) was the one used for spectral acquisition. Raw FT-IR spectra were converted in absorbance. The measurements were performed in the range of 600–4000 cm^−1^ with 120 scans at a resolution of 4 cm^−1^ optical resolution. The averaged spectra were obtained using 32 scans, including subtraction of a background scan of the clean diamond crystal. The diamond was cleaned between samples using alcohol. The spectral data were compared with reference to identifying the functional groups existing in each sample. 

### 2.5. Cancer Cell Models

Human liver (HepG2) cell lines were from American Type Culture Collection (Manassas, VA, USA). Cells were cultured in low-glucose Dulbecco’s Modified Eagle Medium (DMEM) supplemented with 10% fetal bovine serum and 1% penicillin/streptomycin and incubated at 37 °C, in a humidified atmosphere of 5% CO_2_.

#### HepG2 Culture Conditions and Treatments

After a few passages, HepG2 cells were plated at a density of 10,000 cells/mL (viability and cytoprotection assays), 30,000 cells/mL (determination of NO levels), 20,000 cells/mL (comet assay) and 75,000 cells/mL [nuclear staining and reactive oxygen species (ROS) production]. After 24 h, different concentrations of cherry fractions dissolved in medium containing 0.5% (*v/v*) dimethylsulfoxide (DMSO) (6.25–100 µg/mL) were added. To assess the cytotoxicity effects after exposure to the fractions, plates were incubated for 24, 48 and 72 h [28,29,30,31,32]. On the other hand, for evaluating the antioxidant potential, preliminary assays were performed to choose the appropriate concentration and exposure time of each oxidative stress inductor able to cause around 50% cell death (data not shown). Then, cells were first exposed to the fractions for 24 h. After that, the fractions were removed, and they were exposed to 0.5, 1, 1.5 and 2 mM SNP for 24 h, or 200 µM H_2_O_2_ for 6, 12 and 24 h [29,33]. Regarding the determination of NO levels, cells were exposed to 1 and 2 mM SNP for 24, 48 and 72 h, respectively [29]. 

Culture conditions and procedures were common through all assays. All studies were conducted when the cells were in the logarithmic growth phase.

### 2.6. MTT Reduction

After cell treatment, the culture medium was discarded, and the adherent cells were treated for 4 h with 0.5 mg/mL MTT. Then, the MTT was removed, and formazan crystals were dissolved with 100 µL DMSO, including the control group, and quantified by measuring the absorbance at 570 nm in a microplate reader Bio-Rad Xmark spectrophotometer (Bio-Rad Laboratories, Hercules, CA, USA) [28]. Cell viability was expressed as a percentage of the control. Six experiments were performed in triplicate.

### 2.7. Membrane Integrity Assay

The release of the stable cytosolic enzyme lactate dehydrogenase (LDH) into the medium by cells with disrupted cell membrane was spectrophotometrically determined at 340 nm (Bio-Rad Laboratories, Hercules, USA), in a kinetic mode, by following β-nicotinamide adenine dinucleotide (NADH) oxidation during the conversion of pyruvate to lactate. Briefly, after cell treatments, 50 µL of culture medium was placed in 96-well plates, together with 200 µL of NADH (252.84 mM) and 25 µL of pyruvate (14.99 mM) [12]. Both pyruvate and NADH solutions were prepared in phosphate-buffered saline (PBS) (pH 7.4). A decrease in absorbance is directly related to the quantity of LDH released by the cells in the culture environment. A total of six independent experiments were performed. Untreated cells were used as a control.

### 2.8. Cells’ Morphology

Nuclear staining with 4,6-diamidino-2-phenylindole (DAPI) was executed after 24, 48 and 72 h of incubation with the fractions at the indicated concentrations. Briefly, treated cells were washed with PBS and fixed with 4% of paraformaldehyde (PFA) in PBS for 10 min at room temperature. Then, fixed cells were washed again with PBS and incubated with 1 µg/mL DAPI solution for 10 min at room temperature, protected from light. After that, cells were washed two times with PBS and chromatin fluorescence was analysed using a Zeiss AxioImager A1 fluorescence microscope with 405 nm laser excitation for DAPI. Digital images were generated with Zeiss ZEN software [30].

### 2.9. Determination of DNA Damage (Comet Assay)

After exposure to the extracts, cells were collected by trypsinization, and DNA single strand-breaks were evaluated by the alkaline version of the comet assay based on the work of Tulipani et al. [32] with some modifications. Basically, after trypsinization, treated and nontreated cells were suspended in 1% low melting point agarose dissolved in ultra-pure water (37 °C) and immediately pipetted onto frosted glass microscopic slides pre-coated with a layer of 1% (*w/v*) normal melting point agarose also prepared in PBS. Without delay, glass cover slips were placed on the top of the slides, and the agarose/cell mixture was put at 4 °C for 5 min. After that, the cover slips were removed and slides were immersed in ice-cold lysis solution [2.5 M NaCl, 0.1 M EDTA, 10 mM Tris (pH 10) adjusted to pH 10 with 10 M NaOH and supplemented with 1% (*v/v*) Triton X-100] for 16 h to remove cell proteins. Next, slides were placed in a submarine gel electrophoresis for 30 min to unwind before being electrophoresed at 20V (300 mA) for another 30 min. Following electrophoresis, the slides were firstly immersed in neutralizing buffer (0.4 M Tris-HCl, PH 7.5, 4 °C) for 10 min, and then in ultra-pure water for another 10 min. After neutralisation and fixation, slides were stained with DAPI (1 µg/mL) and covered with coverslips for 15 min to colour the DNA, before being visualised using a Zeiss AxioImager A1 fluorescence microscope with 405 nm laser excitation for DAPI. Digital images were generated with Zeiss ZEN software. One hundred nucleoids per slide were scored visually for comet tail size based on an arbitrary scale of 0 (undamaged nucleus) to 4 (extensive damage of DNA). Samples were scored blindly, and results were expressed as arbitrary units.

### 2.10. Determination of NO Levels

The quantity of NO in cell culture medium was determined by its conversion to nitrite, using a mixture of 75 µL of culture media with an equal volume of Griess reagent (1% sulphanilamide and 0.1% N-[naphth-1-yl]ethylenediamine dihydrochloride in 2% H_3_PO_4_). The 96 well-plate was incubated during 10 min at room temperature in the dark. Then, the absorbance was measured at 560 nm in a microplate reader (Bio-Rad Laboratories, Hercules, USA) [29]. The results correspond to the mean ± SEM of 5 independent experiments performed in triplicate.

### 2.11. Nitric Oxide (NO) Scavenging Assay

The ability of cherry fractions in capturing NO was based on the work of Jesus et al. [34]. Briefly, 6 different concentrations were prepared. Then, each well was composed of 100 µL of each extract dissolved in potassium PBS (100 mM, pH 7.4) and 100 µL of SNP (20 mM). On the other hand, blank and control contained 100 µL of phosphate buffer and 100 µL of sodium nitroprusside dihydrate. The plates were incubated at room temperature for one hour under light. Then, 100 µL of Griess reagent (1% sulfanilamide and 0.1% naphthylethylenediamine in 2% H_3_PO_4_) was added to each well and incubated for 10 min in the dark (blanks received 100 µL of H_3_PO_4_). After this time, the absorbance was read at 560 nm. Three experiments for each extract were performed in triplicate.

### 2.12. Measurement of Intracellular Reactive Oxygen Species

The measurement of intracellular ROS was monitored using the fluorescent probe, 2,7-dihydrodichlorofluorescein diacetate (DCFH-DA) according to a previously described method [31], with some modifications. After cells exposure to the fractions, the medium was removed, and the cells were incubated with 200 µL of DCFH-DA for 15 min at 37 °C. After that time, DCFH-DA was discarded, and the cells were incubated with 200 µL of DAPI dissolved in PBS (1:1000). Positive control was incubated with 75 µL H_2_O_2_ 30% for 20 min at 37 °C. After 2 washes with incomplete DMEM, intracellular ROS levels were immediately visualised under a confocal microscope LSM 710 (Carl Zeiss). Five independent experiments were performed in triplicate. 

### 2.13. Preparation of Protein and Ligands

To carry out this study, the 3-dimensional structure of i-NOS protein (Human self-inducible nitric oxide synthase, PDB: 2NSI) was selected and downloaded from the Protein Data Bank (http://www.rcsb.org//pdb (accessed on 15 September 2021)) and prepared by removing, when necessary, the ligand and water molecules, using BIOVIA Discovery Studio Visualizer 2020. After, the molecule was saved in pdb format. To perform the protein-ligand interaction studies, the most abundant phenolic compounds in each fraction were selected and the 3-dimensional structure of the ligands was downloaded from Pubchem (http://pubchem.ncbi.nlm.nih.gov (accessed on 15 September 2021)) in sdf format and converted into pdb by BIOVIA Discovery Studio Visualizer 2020. The PubChem CID for each compound is the following: 3-*O*-caffeoylquinic acid, 1,794,427; caffeic acid, 689,043; cyanidin 3-*O*-glucoside, 441,667; cyanidin 3-*O*-rutinoside, 441,674; kaempferol 3-*O*-rutinoside, 5,318,767; *p*-coumaric acid, 637,542; quercetin, 5,280,343; quercetin 3-*O*-glucoside, 5,280,804. Additionally, the properties of the active compounds were calculated using Lipinski’s rule of five calculated on the SWISSADME predictor (http://www.swissadme.ch/ (accessed on 15 September 2021)).

### 2.14. Receptor-Ligand Docking

Docking was performed using AMDocking [35], with Autodock Vina (AV) [36], which is a graphical tool for molecular docking. Autodock Vina (AV) was run following common protocols. Grind box was delimited manually based on the position of standard inhibitor ethylisothiourea, using PyMol software [37] and the obtained coordinate was added to AV software to define grind box, adapted to the optimum size for each ligand. The resulting docking solutions were subsequently clustered with a root-mean-square deviation (rmsd) tolerance of 2.0 Å and were ranked by binding energy values. In addition, Charmm Force Field (v 1.02) was employed to determine binding energy with 10.0 Å as a non-bonded cut-off distance and distance-dependent dielectric. The lowest binding energy conformer was searched out of ten different conformers for each docking simulation in each delimited grind box. The results of the docking calculation were shown in the output in notepad format. The ligands’ docking conformation was determined by selecting the pose with the highest affinity (most negative Gibbs’ free energy of binding/ΔG). Docking results were visualized with BIOVIA Discovery Studio Visualizer 2020.

### 2.15. Inhibition of Lipid Peroxidation

The inhibition of lipid peroxidation was assessed according to previous methods [38,39]. Briefly, tissue samples of porcine (*Sus scrofa*) liver were homogenized with cold Tris-HCl (10 mM; pH 7.2), to produce a 1:10 (*w/v*) hepatic tissue homogenate. After centrifugation, 100 µL of the supernatant was incubated with 200 µL different concentrations of each fraction for 1 h at 37 °C. Afterward, 500 µL trichloroacetic acid (28% *w/v*) and 380 µL thiobarbituric acid (1%, *w/v*) were added, and the mixture was heated at 100 °C for 15 min. After centrifugation at 3000× *g* for 10 min to remove the precipitated protein, the colour intensity of the malondialdehyde-thiobarbituric acid was read at 532 nm. The inhibition ratio (%) was calculated using the following formula: [(A/B)/A] * 100%, where A and B were the absorbance of the control and the sample solution, respectively. Three experiments were performed in triplicate.

### 2.16. Ferric Reducing-Antioxidant Power Assay

The ferric reducing-antioxidant power assay in hepatic tissue samples was performed according to the colorimetric method described by Benzie and Strain [40], with slight modifications. In brief, tissue samples of porcine (*Sus scrofa*) liver were homogenised in PBS (pH 7.4) to produce a 1:10 (*w/v*) hepatic tissue homogenate. After centrifugation, 9 µL of the supernatant was incubated with 27 µL different concentrations of each fraction and 270 µL of FRAP reagent (containing Fe^3+^-2,4,6-tripyridyl-s-triazine (TPTZ), iron trichloride hexahydrate and acetate buffer at pH 3.6). The antioxidant potential of the samples was determined by monitoring the changes in absorbance at 593 nm due to the reduction of the Fe^3+^-2,4,6-tripyridyl-s-triazine (TPTZ) complex to a coloured Fe^2+^-TPTZ complex induced by the samples, after 40 min of incubation at 37 °C. Three experiments were performed in triplicate.

### 2.17. Statistical Analysis of Results

Statistical analysis was performed using Graphpad Prism Version 8.4.3 (San Diego, CA, USA). One-way ANOVA followed by Dunnett’s test as a post-hoc test that was used to determine the statistical significance in comparison to control. Values of *p* < 0.05 were considered statistically significant. 

## 3. Results and Discussion

### 3.1. FT-IR Spectroscopy Analysis

FT-IR analysis is largely used for being simple to perform, economical, as well as due to their possibility to provide a molecular fingerprint of the sample by featuring their molecular vibrations (stretching, bending, and torsions of the chemical bonds), and thus allowing detailed analysis of the characteristics of the samples [41]. Therefore, the identification of the extracted compounds was performed based on the different absorption spectra registered according to the different types of chemical bonds and their functional groups. Both fractions and total extract showed similar absorbance bands, which was predictable considering they are enriched in several phenolics composed by a benzene ring and carboxyl and hydroxyl groups. The main differences are obtained in the value of the transmittance and the appearance of the bands, namely width and sharpness (data not shown). 

Chlorogenic, caffeic, p-coumaric, 3–4 hydroxybenzoic and 3-OH-hydroxybenzoic acids, kaempferol, kaempferol 3-*O*-rutinoside, quercetin, cyanidin 3-*O*-glucoside, cyanidin 3-*O*-rutinoside and pelargonidin-3-*O*-rutinoside were selected to exhibit the mid-infrared spectral profile of phenolics. For homology, it was possible to detect the presence of many phenolics already reported in sweet cherries, such as hydroxybenzoic, chlorogenic, caffeic and p-coumaric acids, and quercetin and kaempferol derivatives in non-coloured extract and total extract, and cyanidin and pelargonidin glycosides on coloured fraction and total extract. The obtained data are in accordance with the previous studies of the research group, which analysed these fractions by chromatographic techniques and revealed their richness in hydroxycinnamic acids (total extract and non-coloured fraction) and cyanidin derivatives (total extract and coloured fraction) [12,27].

The FT-IR spectra of cherry fractions showed a peak at ~3350 cm^−1^, which indicates a stretching vibration of O-H group, and hence, the presence of carboxyl groups, phenols or amino acids, as reported by previous studies [41,42,43,44]. Furthermore, the absorption band at 3000 cm^−1^ results from vibrations of C-H, which suggests the presence of the benzene ring [45]. On the other hand, the spectral region between 1800 and 1200 cm^−1^ is considered a characteristic of phenolics, being associated with C=O and C=C bond vibrations and aromatic ring deformations observed in methyl, methylene and hydroxyl groups of flavonoids [46]. Particularly, the peak around 1640 cm^−1^ is due to O-H vibrations and aromatic C=C and C=OO^-^ strengths that occur on flavonoids structure [41,47,48]. Moreover, the deformation vibration of the C-C bonds in phenolic groups adsorb in the region between 1500 and 1400 cm^−1^ [49]. In fact, the peak observed around 1500 cm^−1^ is associated with the stretching of aromatic C=C existing on aromatic rings [44,50]. On the contrary, the band at 1400 cm^−1^ is commonly associated with the stretching vibration of methyl, methylene, flavonoids and aromatic rings [43], while the peak around 1300 cm^−1^ is closely attributed to the O-H groups of phenols [49]. Additionally, the peaks around 1200 and 1000 cm^−1^ are often related to C-O stretching and the vibration of C-OH groups from alcohols, ethers, esters, and carboxylic acids of hydroxyflavonoids and phenolic acids [47]. The maximum absorption peaks are identified in the frequency range of 1100–1000 cm^−1^ are synonymous with fluctuations related to the C-O groups of phenolics [46]. The absorption peaks that ranged from 900 to 700 cm^−1^ resulted from the out-of-plane deformation vibrations of C-H in benzene rings [50]. The other peaks, starting around 700–600 cm^−1^ were not identified. 

The obtained data suggest that phenolic-enriched fractions from sweet cherries present many functional groups, which, in turn, have already been demonstrated to possess notable antioxidant, antimutagenic and anti-cancer activities. Similar bounds were found in other studies focused on cherries functional groups [44,46,51].

### 3.2. Effect of Sweet Cherry Fractions in HepG2 Cell Viability

Bioactive compounds from natural products have been a target of many studies owing to their broad spectrum of therapeutic properties with minimal side effects. Focusing on cancer, phenolics have already proved the ability to interfere with the various cancer growth and development stages and, therefore, it is not surprising that they are considered promising chemotherapeutic and/or cancer adjuvant agents [7,9,10,12,14].

All cherry extracts revealed the ability to reduce the mitochondrial activity of HepG2 cells in a dose-dependent manner (Figure 2), standing out the effect of the coloured fraction after 72 h of treatment, which revealed an IC_50_ value of 27.24 ± 0.72 µg/mL. As expected, the most significant LDH response was also obtained using the coloured fraction after 72 h of exposition in a dose-dependent manner, mainly for the highest tested concentrations (25, 50 and 100 µg/mL), with values of 104.7%, 106.2% and 120.2%, respectively. These concentrations were already tested on normal human dermal fibroblast (NHDF) and Madin–Darby canine kidney (MDCK) cells and did not show any cytotoxic effects (cells viability > 80%), which is another evidence about the selective toxicity to HepG2 cells (data not published yet).

Compiling MTT reduction and LDH leakage results, it was clear that MTT results were more expressive. Both data suggest that the loss of mitochondrial activity happened before the membrane’s damage, ruling out a necrotic process in the lowest concentrations (6.25–25 µg/mL) and its occurrence in the highest concentrations tested, which were accompanied by an increase of LDH in the culture medium (50 and 100 µg/mL). In agreement with these results, it was observed higher amounts of debris (Figure 3), as well as cells shrinkage events, and consequently, cytoplasmic blebs and cell structure losses as the concentration increased (Figure 4). 

As expected, and considering the data of the previous Figure, an increase in debris was observed in a concentration-dependent manner. 

Marked morphological alterations, including cells shrinkage, and consequent structure losses, cell death and cytoplasmic blebs, were clearly found. 

To understand the extensive damage in DNA, the comet assay was achieved, and it clearly shows the cytotoxicity effects of the coloured fraction (Figure 5). Control cells presented a genetic damage index of 18 ± 0.5, whereas cells treated with 25, 50 and 100 µg/mL coloured fraction had a comet score of 165 ± 0.4, 202 ± 0.4 and 322 ± 0.4, respectively. This obtained data indicate that the DNA damage becomes more extensive as extract concentration increases. The obtained data are in accordance with other studies [52,53,54].

In a general way, the obtained data were expected since it was already described that phenolics and phenolic-enriched extracts, namely in anthocyanins, can affect cell functions and induce toxicity effects on various cancer cell lines. Recently, Xiao et al. [55] revealed that the cyanidin 3-*O*-rutinoside can induce apoptosis on Hepg2 cells (IC_50_ value of 22.62 µg/mL). Additionally, Gonçalves et al. [12] already reported that anthocyanins-enriched fractions from sweet cherries were more effective in interfering with Caco-2 cells growth than the non-coloured phenolics fraction and total extract. Focusing on HepG2 cells, Forbes-Hernández et al. [28] revealed that anthocyanin-enriched fractions of strawberries exhibited more notorious effects to affect the viability of these cancer cells than their whole methanolic extracts. Furthermore, anthocyanins from *Vitis coignetiae* Pulliat (5 µg/g of animal per day) already showed the capacity to reduce tumour development in mice infected with Hep3B human hepatocellular carcinoma cells as compared to the control group [56]. Similar results were verified after the daily administration of anthocyanins extract from *Lonicera caerulea* fruits (200 mg/kg bw/day) in hepatoma cells (H22)-bearing mice after 15 days of treatment [57]. 

The strong anti-cancer effects of anthocyanins are intimately linked to their catechol, pyrogallol, and methoxy groups present in their chemical structure, which confer them notable antioxidant, anti-cancer, antimutagenic and anti-inflammatory effects, including the capacity to (i) interfere with ERK, JNK, MAPK, NF-κB and PI3K/Akt pathways, (ii) arrest cells proliferation in G1/S and G2/M phases, (iii) activate the caspase cascade, (iv) increase intracellular antioxidant enzymes, (v) modulate aromatase activity, (vi) regulate estrogenic/antiestrogenic levels, (vii) reduce mitochondria membrane potential and (viii) inhibit procarcinogens activation by interfering with phase I metabolizing enzymes, such as cytochrome P450 [11,56,57,58,59,60].

### 3.3. Cell Culture Radical Scavenging Activity

Considering that oxidative stress contributes to the development of many disorders, including cancer, the second step of this study was to evaluate the capacity of the enriched-phenolic fractions and total extract from sweet cherries to protect HepG2 cells against induced-oxidative stress.

In a first step, it was verified that, the preincubation with the highest concentration tested (100 µg/mL), of both fractions and total extract revealed capacity to reduce ROS concentration (Figure 6) near basal levels. The coloured fraction was the most effective to improve the redox intracellular status in cells under induced oxidative stress, with respect to positive control (100%).

Among ROS, H_2_O_2_ is a well-known ROS genotoxic agent whose overproduction and consequent accumulation induces severe oxidative DNA damage, e.g., DNA strand breakage and base modification and lipid peroxidation [33]. Having the previous findings in mind, we also tested the ability of phenolic-enriched fractions and total extract to protect HepG2 cells against H_2_O_2_. Cells were then treated with different concentrations (6.25–100 µ/mL) of each fraction 24 h prior to H_2_O_2_ exposure (200 µM for 6, 12 and 24 h). Cellular viability was again determined by MTT and LDH leakage assays. The obtained data revealed that both phenolic-enriched fractions as well as the total extract were safe at the therapeutic level and had a significant ability to prevent oxidative stress. In fact, all the extracts exerted dose- and time-dependent protective effects in the MTT reduction and LDH leakage assays (Figure 7). As expected, the highest concentration tested (100 µg/mL) was the one that showed the most pronounced protective effects. Among fractions, the non-coloured was more effective than the other ones, revealing increments on cells viability of 19.04% (after 24 h of exposure), comparatively with 6 h of treatment with the same concentration. On the other hand, a slight reduction in cells viability was observed after 24 h of exposure with the coloured fraction and total extract (viability of 69.31 and 74.95% against 73.34 and 76.89%, respectively, obtained after 6 h of treatment (concentration of 100 µg/mL). This reduction was not surprising and can be associated with the presence of anthocyanins, whose pro-oxidant behaviour and easy capacity to be transformed into radicals were already known [61,62]. 

Additionally, we also assessed the ability of each fraction and total extract to quench NO. These radicals are extremely reactive with other pro-oxidant species, causing severe damage in cells membrane. The generation of these radicals was promoted by SNP [29].

Firstly, and in order to understand the possible protective effects of the tested concentrations, cells were treated with each fraction or total extract for 24 h and then subjected to SNP-induced toxicity at concentrations of 0.5, 1, 1.5 and 2 mM for 24 h (Appendix A). Once all concentrations of each fraction showed protective effects, we decided to evaluate their NO scavenging capacity in cell culture medium, at different times (24, 48 and 72 h) and SNP concentrations (1 and 2 mM). As expected, all fractions showed the capacity to reduce NO levels in a dose- and time-dependent manner (Figure 8A–F). In the majority of the conditions, the scavenging activity was more marked using 2 mM of SNP, and the highest concentration tested (100 µg/mL) in all fractions was the most effective to scavenging NO in cells medium. Moreover, a significant reduction in NO levels was observed between 24 and 48 h of SNP exposure, independently of its concentration, being similar between 48 and 72 h. Even so, a notorious reduction was verified using the coloured fraction after 48 h of exposure, with 2 mM SNP (NO reduction of 54.64% vs positive control). However, and in accordance with the findings obtained in the previous assay with H_2_O_2_, it was also verified pro-oxidant effects on fractions with anthocyanins, i.e., total extract and coloured fraction, resulting in NO levels increments of around 4% after 72 h of exposure comparatively with those verified at 48 h. This happens due to the oxidation generated by H_2_O_2_, which in turn, results in the formation of superoxide radicals and H_2_O_2_ species [63,64].

This reduction in NO levels can be due to the capacity of phenolic-enriched fractions from sweet cherries to modulate iNOS and/or the ability to scavenge NO (antioxidant effect), we decided to investigate if a process of direct NO scavenging could be taken place, also contributing to diminishing NO levels in the cellular assay. For this reason, two different assays were executed. The first one was a cell-free assay based on NO scavenging capacity, using the same concentrations studied in the cell system. Once again, SNP was utilized as NO donor. 

### 3.4. NO Scavenging Activity

Overall, all fractions displayed a moderate ability to capture NO (Appendix A). Even so, the coloured fraction was the most active, leading to a reduction of 38.93% in NO levels vs control at the highest concentration tested (100 µg/mL), followed by the total extract (−22.22%) and non-coloured fraction (−11.55%). At the same time, we decided to perform molecular docking studies. 

### 3.5. Rule of Five (Ro5)

Before performing docking, it is important to note that some authors refer caution in using it to design efficient drugs and appeal to its use together with other computation studies, such as Lipinski´s rule of five [65,66]. Therefore, Lipinski’s rule of five (Ro5) of the docking compounds calculated on the SWISSADME predictor was firstly performed, and the obtained results are shown in Table 1. Four of the eight compounds studied complied with Lipinski’s rule of five, presenting a good gastrointestinal diffusion and absorption capacity [67]. However, some drugs, although violating the rule of 5, present a good gastrointestinal permeability [68]. This fact is due to issues of permeability and extensive metabolization along the gastrointestinal tract, which, in turn, enhances their absorption [67,68]. In accordance with the mentioned, several studies based on in vitro and in vivo assays already demonstrated that phenolics present higher bioavailability than expected, including those present in sweet cherries; for example, the bioavailability of phenolic acids varies between 8 and 72%, while that of flavonols from 12 to 41%, 2 to 8% for flavan-3-ols, and between 20 and 55% for anthocyanins [69,70,71].

### 3.6. Docking Results

The results obtained by computing docking using AMDocking [35] with AV [36] show that some of the compounds used have binding energies highly compatible with inhibition by competition with the active site of the i-NOS enzyme, namely cyanidin 3-*O*-rutinoside (−11.4 kcal/mol), kaempferol 3-*O*-rutinoside (−10.8 kcal / mol) and cyanidin 3-*O*-glucoside (−10.1 kcal/mol) (Table 1). Likewise, these compounds are those that present greater interaction surfaces with the active site (Figure 9). Even so, other compounds also appear to have outstanding potentials, such as quercetin, that displayed a remarkable interaction surface with computational values of binding energy ΔG of −9.4 kcal/mol and concentrations inhibitory score of 0.128 nM. However, the three compounds that had shown the best binding energy were those with the lowest inhibitory concentration (estimated Ki): cyanidin 3-*O*-rutinoside (0.005 nM), kaempferol 3-*O*-rutinoside (0.012 nM), cyanidin 3-*O*-glucoside (0.038 nM) and *p*-coumaric acid (0.006 nM). This last compound also exhibited the highest ligand efficiency (−0.59), which was not surprising since this parameter was higher in smaller size molecules. 

In this regard and taking into account the depending on the nature of the molecule, we observe all compounds present a similar interaction surface with the active site, with similar morphology and variations in the potential of hydrogen bonds established with the receptor (Figure 9A–H). Even so, according to the 2D receptor-ligand interaction diagrams (Figure 9I–P), the residues PHE:369 and TRP:194 are those that have a more determining role in the interaction with these compounds. Although other authors have already highlighted that molecules with cis-fused rings have more notorious activity against the active group, allowing them to internalize the active site [72]. In this study, the compounds that exhibited the most complex 3D structures, given their rings arrangement and substituents, were the ones that presented more complex interaction structures in the active site with lower binding energies. This fact is closely related to the complexity of the established surfaces. Additionally, it should be noted that some phenolic compounds extracted from *Polygonum orientale* L., showed similar interaction values to those obtained in this work, as was the case of quercetin (−9.41 vs. −9.40) [73].

Given the obtained results, it was possible to conclude that the decrease of cellular NO levels was due to a combination between the interaction of phenolics and iNOS together with their ability to capture NO.

The obtained findings agree with previous studies [12,61,62]. In fact, enriched-phenolic fractions already displayed the capacity to relieve oxidative stress [12]. These abilities are due to their chemical structure, standing out their multiple hydroxyl groups, which are responsible for neutralizing free radical species and chelate metals [74]. However, sometimes, several substitutions by hydroxyl groups, namely in the B ring, can significantly increase the pro-oxidant behaviour of flavonoids, which enhance cellular reactive species concentrations and also intensify their cytotoxic levels, killing cancer cells [63,64,74]. Of course, this pro-oxidant behaviour is also dependent on the concentration used [75]. Therefore, the mentioned evidence explains the fact that the coloured fraction and total extract slightly reduce their antioxidant potential in some induced-oxidative stress conditions at the highest tested concentrations. 

### 3.7. Protective Effects of Sweet Cherry Fractions against Lipid Peroxidation and Ferric Species

Additionally, the capacity of phenolic-enriched fractions from sweet cherries to protect homogenates of the porcine liver was also evaluated. As far as we know, this is the first time that both assays were performed using liver tissues and phenolics extracted from cherry fruits. 

Therefore, lipid peroxidation was indirectly measured based on the capacity of the extracts to inhibit the formation of thiobarbituric acid-reactive substances (TBARS). In a general way, both fractions and total extract showed capacity to protect the porcine liver against this damage (Figure 10A–C). Among fractions, the non-coloured was the most effective, showing capacity to inhibit TBARS formation in a dose-dependent manner, with inhibitory values ranging from 35.40 (6.25 µg/mL) to 58.87% (100 µg/mL). On the other hand, the total extract diminished its antioxidant effects in the highest concentration tested, i.e., 100 µg/mL, while the coloured fraction showed slight pro-oxidant effects as the concentration increased. Once again, the pro-oxidant behaviour was predictable, being mainly attributed to the presence of various hydroxyl groups on anthocyanins B ring [61,63].

Concerning the capacity of the phenolic-enriched fractions to capture ferric species, the obtained results revealed that all fractions and total extract were able to scavenge ferric species in a dose-dependent manner (Figure 10D–F). Contrary to lipid peroxidation, in this assay, the coloured fraction was the one that showed the most notable capacity, with values varying from 48.21 (6.25 µg/mL) to 63.82% (100 µg/mL), followed by the total extract (53.06–59.38% reduction) and non-coloured fraction (52.47–61.38% reduction), respectively.

The obtained results were expectable, another evidence for the antioxidant potential of phenolics, which is, in part, attributed to their chemical structure, standing out the presence of hydroxyl and methoxy groups, and catechol residue, that easily donate neutralise free radicals and oxidative species, and chelate metals [76]. In fact, both capacities are not surprising, being in accordance with other studies [12,34,38,61]. For example, Bastos et al. [38] already reported the capacity of hydromethanolic extracts of cherry fruits to protect brain tissues from lipid peroxidation (IC_50_ value of 1.46 µg/mL). Furthermore, hydroethanolic extracts from vegetal parts, i.e., stems, leaves and flowers, of *Prunus avium* also showed ability to protect human erythrocytes against TBARS, revealing IC_50_ values of 26.20, 70.91 and 292.40 µg/mL, respectively [34]. As far as we know, and although the capacity of several phenolics present in the fractions to capture ferric species was already well-documented [64,77,78]. This is the first study regarding the ability of enriched fractions from cherries to protect the liver from ferric species.

## 4. Conclusions

The present data bring a significant advantage and provides new insights regarding the capacity of phenolic-enriched fractions from sweet cherries to defeat carcinogenesis and metastasis of liver cancer cells and also to improve their redox state, diminishing reactive and nitrogen species generation. This work is further evidence that sweet cherries consumption should be encouraged and sustains their incorporation on dietary supplements and pharmaceutical drugs. However, and considering the pro-oxidant effects of anthocyanins, it is important to highlight that, in some situations of oxidative stress, the use of extracts with a poor concentration of these compounds can be an added value. Notwithstanding, in order to prevent the risk of toxic effects, animal and clinical trials must be conducted to explore the full anti-cancer potential of sweet cherries and their safe dosage.

## Figures and Tables

**Figure 1 foods-10-02623-f001:**
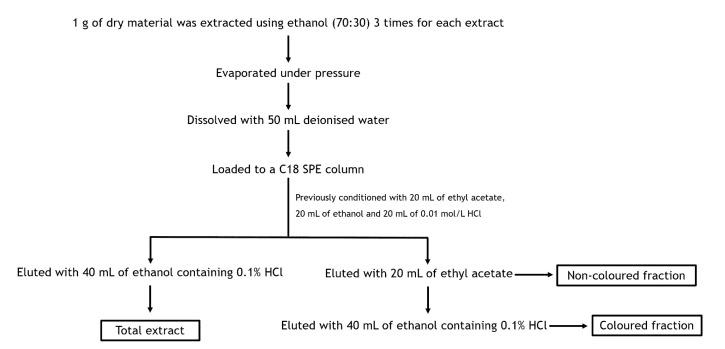
Schematic diagram regarding the obtention of extracts enriched in phenolic compounds. To prevent the degradation of anthocyanins, ethanol was supplemented with 0.1% HCl. The solvent in each fraction was removed by reduced pressure evaporation at 37 °C. Each fraction and total extract were already characterized through chromatographic techniques by our research group. This analysis revealed their richness in hydroxycinnamic acids (69.8% and 99.7% of the total phenolic compounds for total extract and non-coloured fraction, respectively) and cyanidin 3-O-rutinoside (comprising 24.5% and 81.5% of the total compounds for the total extract and coloured fraction) [12,27].

**Figure 2 foods-10-02623-f002:**
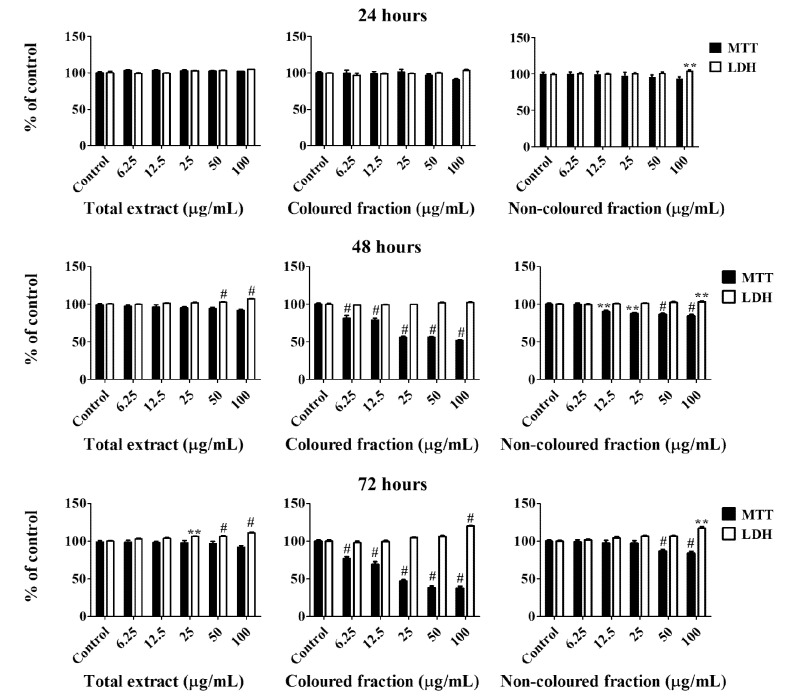
Viability of HepG2 cells assessed by 3-(4,5-dimethylthiazol-2-yl)-2,5-diphenyltetrazolium bromide (MTT) reduction and lactate dehydrogenase (LDH) leakage assays. Cells were treated with each extract for 24, 48 and 72 h. Values show mean ± SEM of six independent assays performed in triplicate compared to the respective control (** *p* < 0.01 and ^#^ *p* < 0.0001).

**Figure 3 foods-10-02623-f003:**
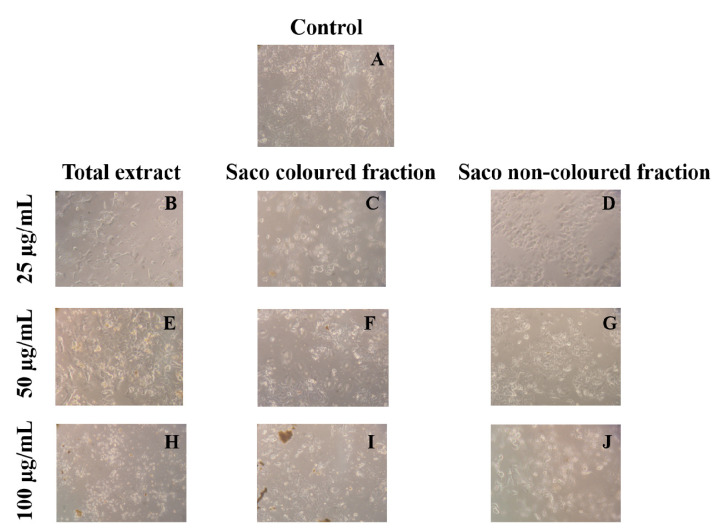
Morphological changes in HepG2 cells (control vs. treatment after 72 h of incubation). (**A**) corresponds to the control, (**B**,**E**,**H**) correspond to Saco total extract, while (**C**,**F**,**I**) to the coloured fraction and (**D**,**G**,**J**) to the non-coloured one, at concentrations of 25, 50 and 100 µg/mL, respectively.

**Figure 4 foods-10-02623-f004:**
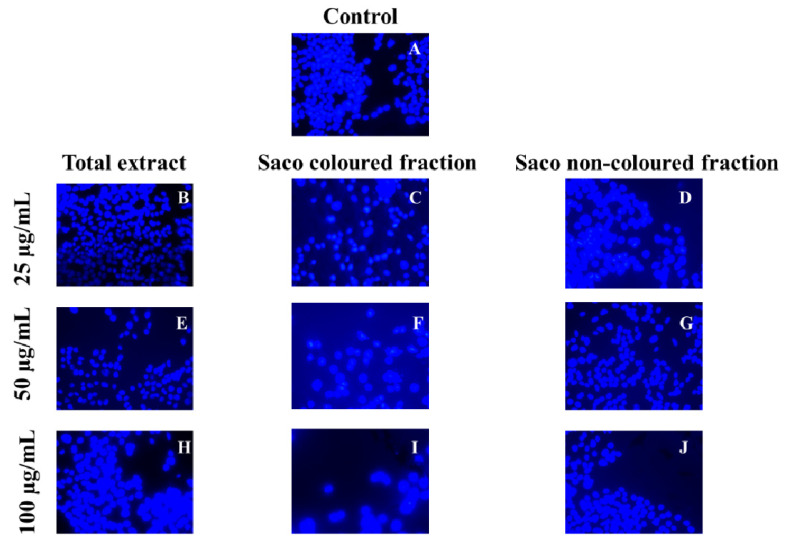
Nuclear morphological study by 4′,6-diamidine-2′-phenylindole dihydrochloride staining of HepG2 cells (control vs. treatment after 72 h of incubation). (**A**) corresponds to the control, (**B**,**E**,**H**) correspond to Saco total extract, while (**C**,**F**,**I**) to the coloured fraction and (**D**,**G**,**J**) to the non-coloured one, at concentrations of 25, 50 and 100 µg/mL, respectively. Three independent experiments were performed, and a representative field was selected.

**Figure 5 foods-10-02623-f005:**
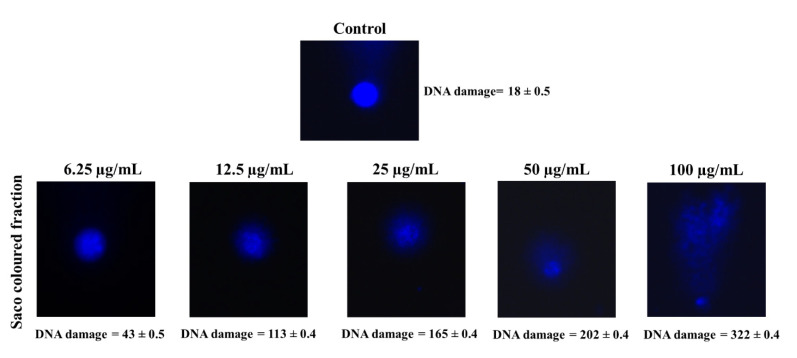
Representative comet images in experiments (control vs. treatment with 6.25, 12.5, 25, 50 and 100 µg/mL the coloured fraction after 72 h of incubation). DNA was stained with 4,6-diamidino-2-phenylindole (DAPI). The number of cells scored in each treatment was 100.

**Figure 6 foods-10-02623-f006:**
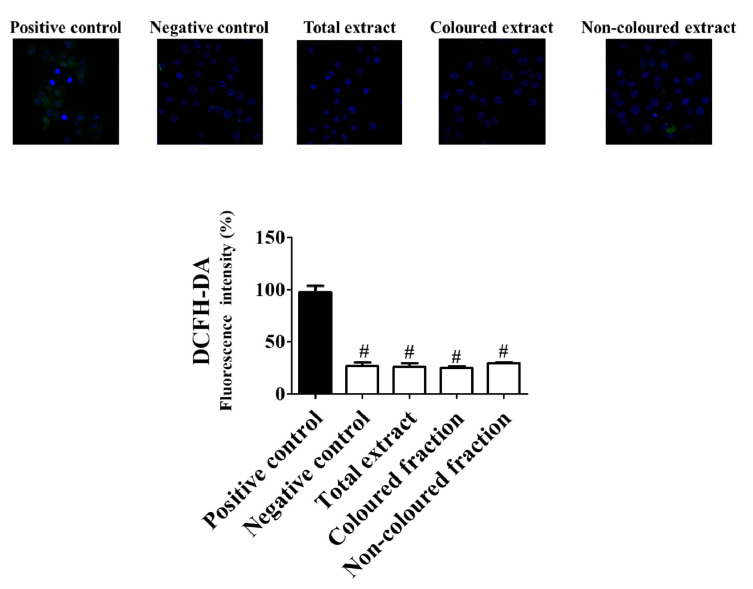
Protective effects of sweet cherry extracts (100 µg/mL) after 24 h of treatment against reactive oxygen species production in HepG2 cells. HepG2 cells were pre-incubated with 2,7-dihydrodichlorofluorescein diacetate (DCFH-DA) (5 µM, 15 min). Control cells were incubated with Dulbecco’s modified Eagle’s medium plus vehicle (negative control) and with 10 mM H_2_O_2_ for 20 min (positive control). All cells were stained with 2,7-dihydrodichlorofluorescein diacetate (DCFH-DA). Results are expressed as mean ± SEM of five independent assays, performed in triplicate (^#^ *p* < 0.0001 compared to the respective positive control). Representative images of HepG2 cells were taken by confocal microscopy.

**Figure 7 foods-10-02623-f007:**
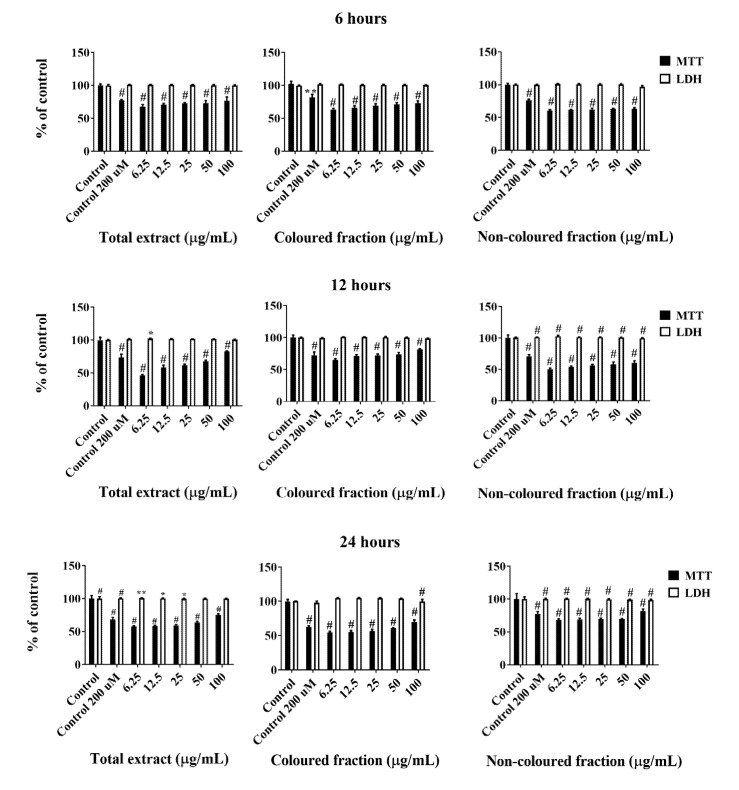
Viability of HepG2 cells assessed by 3-(4,5-dimethylthiazol-2-yl)-2,5-diphenyltetrazolium bromide (MTT) reduction and lactate dehydrogenase (LDH) leakage assays. Cells were pre-treated for 24 h with each extract and then incubated with 200 µM of hydrogen peroxide (H_2_O_2_) for further 6, 12 and 24 h. Values show mean ± SEM of six independent assays performed in triplicate (* *p* < 0.05, ** *p* < 0.01 and ^#^ *p* < 0.0001 compared to the respective control without H_2_O_2_).

**Figure 8 foods-10-02623-f008:**
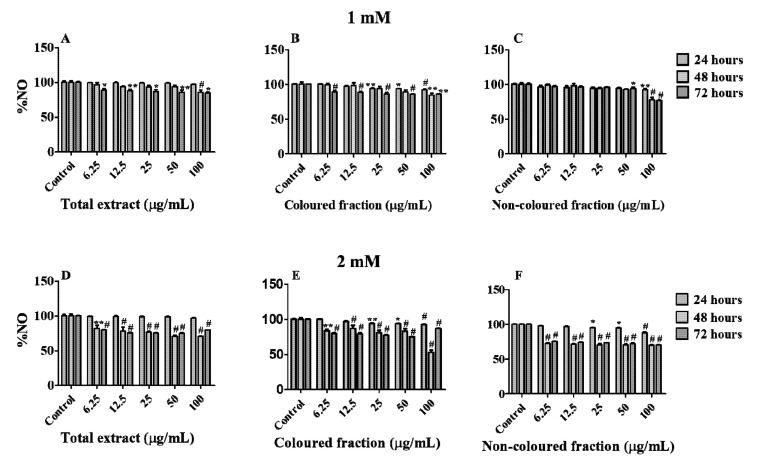
Nitric oxide (NO) levels in HepG2 cells pre-treated for 24 h with each extract and then incubated with 1 mM (**A**–**C**) and 2 mM (**D**–**F**) of sodium nitroprusside for further 24, 48 and 72 h. Values show mean ± SEM of five independent assays performed in triplicate compared to the respective control (* *p* < 0.05, ** *p* < 0.01 and ^#^ *p* < 0.0001).

**Figure 9 foods-10-02623-f009:**
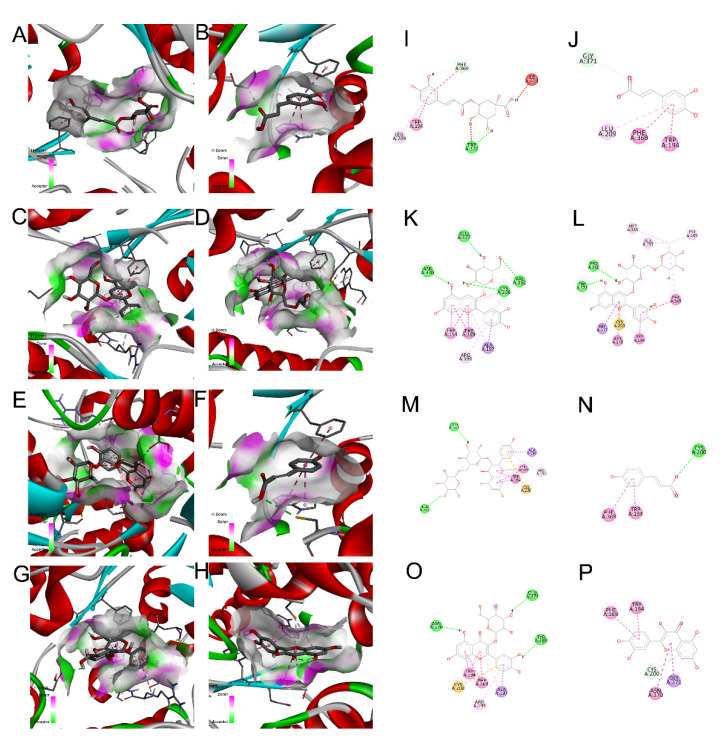
Energy interaction surfaced in a 3D model obtained by docking for each ligand-receptor interaction: (**A**), 3-*O*-caffeoylquinic acid; (**B**), caffeic acid; (**C**), cyanidin 3-*O*-glucoside; (**D**), cyanidin 3-*O*-rutinoside; (**E**), kaempferol 3-*O*-rutinoside; (**F**), *p*-coumaric acid; (**G**), quercetin 3-*O*-glucoside; (**H**), quercetin. 2D model diagram for efficient interaction of each ligand with the receptor: (**I**), 3-O-caffeoylquinic acid; (**J**), caffeic acid; (**K**), cyanidin 3-*O*-glucoside; (**L**), cyanidin 3-*O*-rutinoside; (**M**), kaempferol-3-*O*-rutinoside; (**N**), *p*-coumaric acid; (**O**), quercetin 3-*O*-glucoside; (**P**), quercetin.

**Figure 10 foods-10-02623-f010:**
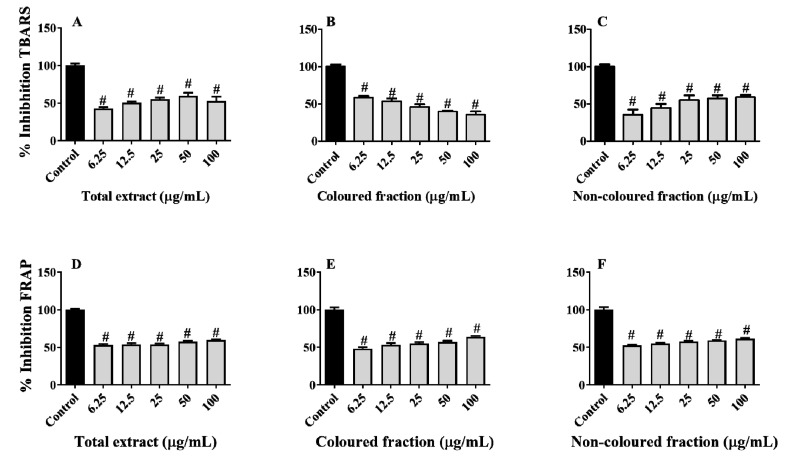
Ability of phenolic-enriched fractions from sweet cherries to inhibit lipid peroxidation (**A**–**C**) and ferric species (**D**–**F**) on tissue samples of porcine (*Sus scrofa*) liver. Data represent the mean ± SEM of three independent experiments performed in triplicate (^#^ *p* < 0.0001 compared to the respective control).

**Table 1 foods-10-02623-t001:** Lipinski’s rule of five (RO5) of i-NOS ligands and molecular docking analysis of its against human self-inducible nitric oxide synthase (2NSI).

Compound	LogP (<5)	H Bond Donor (<5)	H Bond Acceptor (<10)	Violations	Meet Ro5 Criteria	Binding Energy Value ΔG (−kcal/mol)	Estimated Ki (uM)	Ligand Efficiency
3-*O*-Caffeoylquinic acid	−0.83	6	9	1	Yes	−9.1	0.215	−0.36
Caffeic acid	0.25	3	4	0	Yes	−6.8	10.800	−0.52
Cyanidin 3-*O*-glucoside	−2.28	8	11	2	No	−10.1	0.038	−0.32
Cyanidin 3-*O*-rutinoside	−3.24	10	15	3	No	−11.4	0.005	−0.27
Kaempferol 3-*O*-rutinoside	−0.79	9	15	3	No	−10.8	0.012	−0.26
p-Coumaric acid	1.02	2	3	0	Yes	−7.1	0.006	−0.59
Quercetin	0.72	5	7	0	Yes	−9.4	0.128	−0.43
Quercetin 3-*O*-glucoside	−2.30	8	12	2	No	−9.4	0.128	−0.28

## Data Availability

Data are contained within this article.

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
