# Peer review of "Hepatoprotective Effects of Sweet Cherry Extracts (cv. Saco)"

_foods, 2021, doi:10.3390/foods10112623_

Round 1

Reviewer 1 Report

- In Fig.2 24 hours, non colored fraction: is the significance of the MTTvs LDH 25-50-100microg / ml data real?
- In Fig. 3 the morphological changes are not as clear as described by the authors, observation in electron microscopy would be useful to verify real changes.
- in fig. 7 the references of significance are not present in the figure
- in figures 8 and 10 the significance was not calculated.
there are some typos in the text

Author Response

- In Fig.2 24 hours, non colored fraction: is the significance of the MTTvs LDH 25-50-100microg / ml data real?

Authors’ response: Thank you so much for your note, everything was revised and now is correct. We are sorry for the mistake (please see now Figure 2 of the revised version).

- In Fig. 3 the morphological changes are not as clear as described by the authors, observation in electron microscopy would be useful to verify real changes.

Authors’ response: Sorry, we know that you are absolutely right but unfortunately, we are not able to proceed with that by now, since we do not have the cells in culture nor the resources needed. In addition, we would not be able to make it in useful time. We apologize for that. In addition, similar data using this type of methodology was found in other articles. Please see, for example:

  • Ferreres et al. (2015). Pennyroyal and gastrointestinal cells: multi-target protection of phenolic compounds against t-BHPinduced toxicity. RSC Advances (doi: 10.1039/C5RA02710A)
  • Rahman et al. (2013). In vitro morphological assessment of apoptosis induced by antiproliferative constituents from the rhizomes of Curcuma zedoaria. Evidence-Based Complementary and Alternative Medicine (doi: 10.1155/2013/257108).
  • Gonçalves et al. (2020). Multi-target protection of Pterospartum tridentatum phenolic-rich extracts against a wide range of free radical species, antidiabetic activity and effects on human colon carcinoma (Caco-2) cells. Journal of Food Science (doi: 10.1111/1750-3841.15511)
  • Thoda et al. (2020). Profiling of 120 types of herbal extracts and their effects on morphology in cultured neuronal or glial cell lines, followed by RNA extraction for a cDNA library: Consideration for use in studies based on Kampo theories. Traditional & Kampo Medicine (doi: 10.1002/tkm2.1274).
  • Samarghandian et al. (2010). Use of in vitro assays to assess the potential antiproliferative and cytotoxic effects of saffron (Crocus sativus L.) in human lung cancer cell line. Pharmacognosy Magazine (doi: 10.4103/0973-1296.71799).
  • Wang et al. (2017). Antiproliferative and proapoptotic activities of anthocyanin and anthocyanidin extracts from blueberry fruits on B16-F10 melanoma cells. Food & Nutrition Research (doi:10.1080/16546628.2017.1325308)
  • Gonçalves et al. (2018). Antioxidant status, antidiabetic properties and effects on Caco-2 cells of colored and non-colored enriched extracts of sweet cherry fruits. Nutrients, 10(11) 1-20 (doi:10.3390/nu10111688)

- in fig. 7 the references of significance are not present in the figure

Authors’ response: We changed the figures and added the significance, thank you so much for your appointment (please see now Figure 7 of the revised version).

- in figures 8 and 10 the significance was not calculated.

Authors’ response: We changed the figures and added the significance, thank you so much for your appointment (please see now Figures 8 and 10 of the revised version).

there are some typos in the text

Authors’ response: The article was all revised and the typos were corrected. Thank you so much for your note.

Reviewer 2 Report

Dear Editor and Authors,

The manuscript ‘Hepatoprotective effects of sweet cherry extracts (cv. Saco)’ by Ana C. Gonçalves José D. Flores-Félix , Ana R. Costa , Amílcar Falcão, Gilberto Alves, and Luís R. Silva describes research on the effects of three cherry fractions of polyphenols on human hepato cellular carcinoma (HepG2) cells viability, and effectiveness to improve the redox status of these cells under oxidative damage induced by nitric oxide radicals and hydrogen peroxide.

The phenolic  characterization of fractions was performed by Fourier transform infrared spectroscopy with a reference of chromatographic analyzed in previous publications of the group.

Based on the results the Authors found that the phenolic fractions of sweet cherries (cv. Saco, can impair cell viability and suppress cells growth after 72 hours of exposure, promoting necrosis at the highest  tested concentrations (>50 µg/mL). The fractions were also showed capacity to protect these cells against oxidative injury by capturing radicals as well as by modulating reactive oxygen and nitrogen species generation, as demonstrated by bioinformatic  tools.

The manuscript is well written, the methods up to date and well described, references well collected. However, as with all research on cell lines the Authors have not fully considered the possible concentrations of compounds (polyphenols) in body fluids after consumption of sweet cherry, that is bioavailability of the polyphenols. Some words on the topic of bioavailability should be added.

Line 41 there should be ‘was responsible for killing’

Line 51 Therefore, there exists

Line 309 delete ‘de’

Considering all the above I recommend minor revision of the manuscript,

Yours sincerely,

Author Response

The manuscript ‘Hepatoprotective effects of sweet cherry extracts (cv. Saco)’ by Ana C. Gonçalves José D. Flores-Félix , Ana R. Costa , Amílcar Falcão, Gilberto Alves, and Luís R. Silva describes research on the effects of three cherry fractions of polyphenols on human hepato cellular carcinoma (HepG2) cells viability, and effectiveness to improve the redox status of these cells under oxidative damage induced by nitric oxide radicals and hydrogen peroxide.

The phenolic  characterization of fractions was performed by Fourier transform infrared spectroscopy with a reference of chromatographic analyzed in previous publications of the group.

Based on the results the Authors found that the phenolic fractions of sweet cherries (cv. Saco, can impair cell viability and suppress cells growth after 72 hours of exposure, promoting necrosis at the highest tested concentrations (>50 µg/mL). The fractions were also showed capacity to protect these cells against oxidative injury by capturing radicals as well as by modulating reactive oxygen and nitrogen species generation, as demonstrated by bioinformatic tools.

The manuscript is well written, the methods up to date and well described, references well collected. However, as with all research on cell lines the Authors have not fully considered the possible concentrations of compounds (polyphenols) in body fluids after consumption of sweet cherry, that is bioavailability of the polyphenols. Some words on the topic of bioavailability should be added.

Authors’ response: First of all, we would like to thank Your kind comments and compliments made to our manuscript. Following the comments received, the changes made by us are highlighted in the revised version, in accordance with the request. Even so, after each Reviewer’s comment, we indicated the main changes introduced and the corresponding lines.

As recommended, some sentences focused on phenolics’ bioavailability were added (please see now lines 587 to 591 of the revised version).

Line 41 there should be ‘was responsible for killing’

Authors’ response: Thank you so much for your note, the same was corrected (please see now lines 37 to 39 of the revised version).

Line 51 Therefore, there exists

Authors’ response: Thank you so much for your appointment, the same was corrected (please see now line 51 of the revised version).

Line 309 delete ‘de’

Authors’ response: Thank you so much for your appointment, the same was corrected (please see now line 310 of the revised version).

Yours sincerely.